**METHOD**

# Jointly benchmarking small and structural variant calls with vcfdist

Tim Dunn[1*] , Justin M. Zook[2], James M. Holt[3] and Satish Narayanasamy[1]

*Correspondence:
timdunn@umich.edu

[1] Computer Science
and Engineering, University
of Michigan, Ann Arbor,
Michigan, USA
[2] National Institute of Standards
and Technology, Gaithersburg,
Maryland, USA
[3] PacBio, Menlo Park, California,
USA

## Abstract

In this work, we extend vcfdist to be the first variant call benchmarking tool to jointly evaluate phased single-nucleotide polymorphisms (SNPs), small insertions/deletions (INDELs), and structural variants (SVs) for the whole genome. First, we find that a joint evaluation of small and structural variants uniformly reduces measured errors for SNPs (− 28.9%), INDELs (− 19.3%), and SVs (− 52.4%) across three datasets. vcfdist also corrects a common flaw in phasing evaluations, reducing measured flip errors by over 50%. Lastly, we show that vcfdist is more accurate than previously published works and on par with the newest approaches while providing improved result interpretability.

**Keywords:** Benchmarking, Variant calling, Structural variation, Single-nucleotide polymorphism, Insertion, Deletion, Comparison, Phasing

## Background

Prior to the invention of DNA sequencing, structural variants (SVs) larger than 3 Mb were observed using a microscope as early as 1959 [1, 2]. Following the initial sequencing of the human genome in 2001 using short read technologies [3], however, the focus of most research investigations shifted to single-nucleotide polymorphisms (SNPs) and small insertions and deletions (INDELs). It quickly became apparent that SNPs are the most common form of genetic variation, accounting for the approximately 0.1% difference in genomic sequence between two individuals [4], or about 3.1 million SNPs. Short-read technologies were well-poised to investigate these differences, due to their short read lengths but high per-base accuracy. It has since been determined that though SVs and INDELs are less common than SNPs, due to their larger size they account for a further 1.4% difference in genome composition between individuals [5], or about 43.2 million bases.

A few years later, in 2009, the first tools to identify structural variants from short-read alignments were developed [6–8]. Although short-read based structural variant callers remain widely used, they have relatively low recall (10–70%) due to the

inherent difficulties of identifying large insertions and deletions from mapped short reads [9]. The accurate detection of structural variants was greatly assisted by the development of new long-read sequencing technologies around 2014, most notably from Pacific Biosciences (PacBio) and Oxford Nanopore Technologies (ONT) [10, 11]. Although early iterations of each technology had much lower per-base accuracy rates of around 85% [10, 11], longer read lengths led to unambiguous read mappings and more accurate structural variant calls [12]. Since then, accuracy has improved and both PacBio and ONT can sequence reads above 15 Kb with 99 to 99.9% accuracy, rivaling the accuracy of short reads [13, 14]. As shown in Fig. 1, this has led to the recent development of variant calling pipelines built from long-read sequencing data [13].

Once small and structural variants have been called, accurate comparison of variant call files (VCFs) is important for (1) genome-wide association studies (GWAS) [18, 19], (2) precision medicine [20], (3) variant annotation and effect prediction [21, 22], (4) sequencing and variant calling pipeline benchmarking [15, 16], and (5) variant database curation [23, 24]. In short, accurate VCF comparison is necessary for studying the impacts of genetic variants, for understanding the performance of variant calling methods, and for making decisions based on an individual's genetic composition. This information can then be used to identify mutations that cause genetic diseases, to select the best variant calling pipeline for clinical usage, to develop targeted drugs, and to direct future research and funding.

Although small and structural variant calls can now be made from the same analysis pipeline, the current standard practice for VCF benchmarking involves separating small variants (smaller than 50 bp) from structural variants (larger than 50 bp) prior to benchmarking (see Fig. 1). This 50-bp threshold was selected for historical and technical reasons related to the limitations of short-read sequencing, not because a 50-bp threshold is biologically significant in any way [8, 9]. Short-read sequencing's variant calling performance is lower for INDELs larger than 50-bp because the mappability of a 150-bp read containing such a large variant is significantly reduced. For this reason, variants below and above this size threshold have been historically evaluated separately. Prematurely categorizing variant calls into small and structural

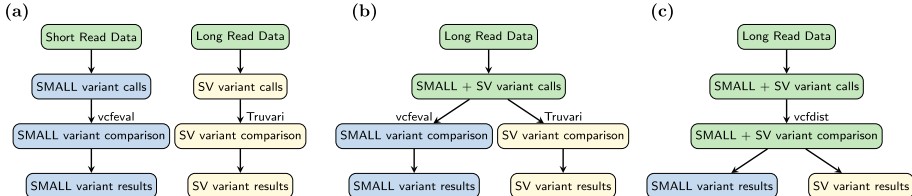

**Fig. 1** **a** Until recently, small and structural variants were called separately using different pipelines because they required different DNA sequencing technologies: short reads cannot be unambiguously mapped to call many structural variants accurately, and long reads were not accurate enough for precise small variant calling. **b** Due to recent improvements in long-read sequencing accuracy, whole genome sequencing (WGS) pipelines can now identify both small and structural variants (SVs) from the same sequencing data. It is still standard practice to evaluate these variant call categories separately, however, using vcfeval [15] for small variants and Truvari bench [16] for large INDELs and SVs. **c** We propose joint benchmarking of small and structural variant calls in this work, by extending vcfdist [17] to evaluate SVs. By comparing query and truth variants across size categories, vcfdist is able to detect a greater number of equivalent truth and query variants. This improves benchmarking accuracy, as shown in Fig. 2

variants prior to benchmarking has a significant impact on measured variant calling performance (see Fig. 2), since several smaller variants are frequently equivalent to one or several larger variants.

The variant call file (VCF) format was first defined in 2011, and a simple exact variant comparison engine was released at the same time as part of vcftools [25]. vcfeval was introduced by Real Time Genomics (RTG) in 2015 and is capable of handling equivalent variant representations [15]. It was designed to evaluate unphased small variant calls, requires exact matches, and evaluates variants up to 1000 bp in size. vcfeval has stood the test of time, being the recommended small variant calling evaluator by the Global Alliance for Genomics and Global Health (GA4GH) in 2019 [26, 27]. In 2023, vcfdist was released to evaluate locally phased small variant calls from long-read sequencing pipelines, relaxing vcfeval's requirement that variants match exactly [17].

Most structural variant calling evaluators similarly allow inexact variant matches. Truvari bench, for example, considers two structural variants equivalent if they are located nearby on the reference, are of similar total size, overlap one another, and have a 70% similar sequence [16]. Although Truvari bench is able to perform whole genome SV comparison, it ignores small INDELs under 50 bp by default and is not currently recommended for evaluating SNPs [16]. Truvari's refine module extends Truvari bench using an alignment algorithm (WFA [28], MAFFT [29], or POA [30]) to harmonize phased

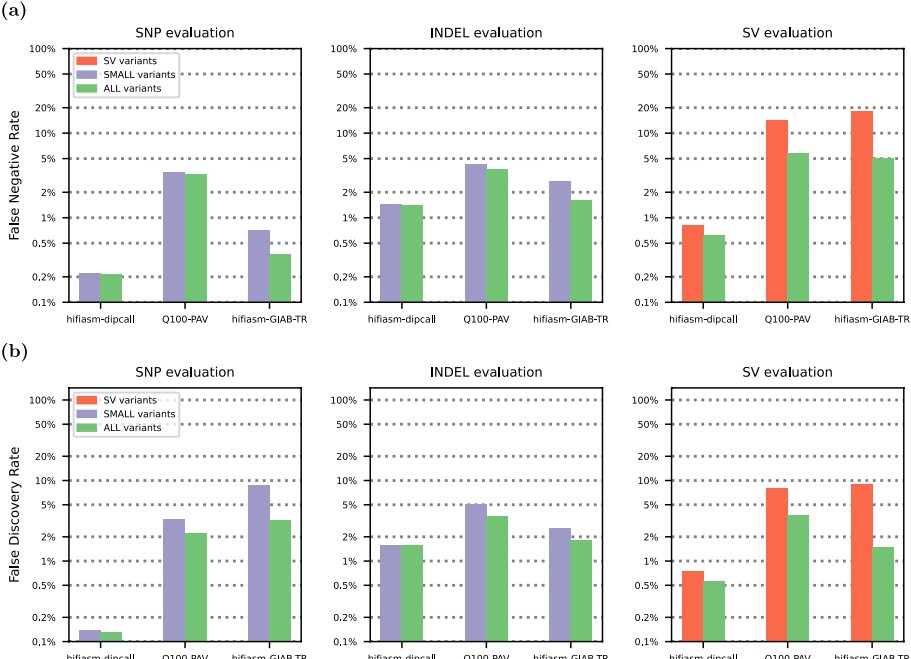

**Fig. 2** We evaluated three phased HG002 whole genome sequencing (WGS) variant callsets (described in Additional File 1: Table S1) on the whole-genome GIAB-Q100 benchmarking BED regions for small and structural variants using vcfdist. We show that compared to existing methodologies, which evaluate small variants (in purple) and structural variants (in red) separately, evaluating all variants at once (in green) leads to higher measured performance for each variant category. **a** False negative rate (FNR) and **b** false discovery rate (FDR) decrease when all variants are evaluated together, across all datasets. This occurs because correctly determining variant equivalence sometimes requires considering variants from multiple categories. Please note that results are plotted on a logarithmic scale

query and truth VCF variant representations for benchmarking INDELs and SVs at least 5 bp in size [31].

In this work, we extend vcfdist to be the first tool to jointly evaluate phased SNP, INDEL, and SV calls for whole-genome datasets. Doing so required major internal restructuring and improvements to vcfdist to overcome scalability issues relating to memory and compute requirements. We show that performing a joint analysis of all variant sizes leads to better measured overall accuracy than when evaluating small and structural variants separately, reducing measured false negative and false positive variant calls by 28.9% for SNPs, by 19.3% for INDELs, and by 52.4% for SVs of over 50 bases. We find that vcfdist's alignment-based analysis obtains more accurate results than vcfeval or Truvari bench and similar accuracy results to Truvari refine, but provides more interpretable results because the representation of evaluated truth and query variants is unchanged. Finally, we jointly evaluate SNP, INDEL, and SV phasing and show that between 42.6% and 92.2% of all phasing flip "errors" that popular phasing analysis tool WhatsHap reports are false positives. Differing variant representations cause variants to appear incorrectly phased, though they are not. These false positive flip errors then lead to false positive switch errors, which will significantly affect downstream tertiary analyses. We demonstrate that vcfdist is able to avoid these errors in phasing analysis by using alignment-based variant comparison.

## Results

### Joint evaluations allow variant matches across size categories and increase measured performance

In order to understand the impact of jointly benchmarking small and structural variants on measured accuracy, we evaluated three whole genome sequencing (WGS) datasets with vcfdist using several different variant subsets from the truth and query VCFs. More information on these WGS datasets can be found in the "Methods" section and Additional File 1: Table S1. Figure 2 shows that compared to existing methodologies, which evaluate small variants (in purple) and structural variants (in red) separately, jointly evaluating all variants (in green) leads to lower measured error rates for each variant category.

In Fig. 2, the hifiasm-dipcall dataset uses alignment parameters which are identical to the draft benchmark Q100-dipcall VCF. As a result, it sees the lowest rates of improvement from a joint evaluation of small and structural variants: a 4.6% reduction in SNP errors, a 1.2% reduction in INDEL errors, and a 24.9% reduction in SV errors. The hifiasm-GIAB-TR VCF uses the same assembly as hifiasm-dipcall with very different alignment parameters and therefore sees great benefits from a joint evaluation: a 62.5% reduction in SNP errors, a 34.8% reduction in INDEL errors, and a 75.3% reduction in SV errors. The Q100-PAV VCF lies somewhere between these two extremes, with a 19.5% reduction in SNP errors, a 21.7% reduction in INDEL errors, and a 57.0% reduction in SV errors. A visualization of the alignment parameters used for each dataset is included in Additional File 1: Fig. S1.

These performance improvements originate from cases where multiple smaller variants are found to be nearly or exactly equivalent to one or several larger variants. Figure 3 shows an example where this occurs and a joint evaluation of small and structural

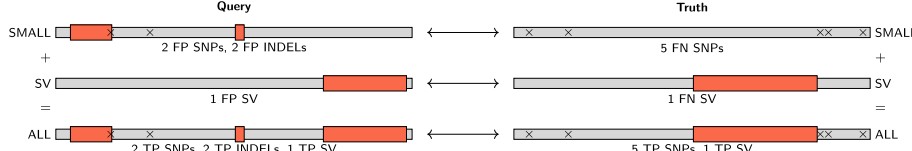

**Fig. 3** An example from the Q100-PAV HG002 variant callset (`chr1:3,287,250–3,287,700` on GRCh38) where using vcfdist to jointly evaluate small and structural variants improves measured performance. Single-nucleotide polymorphisms (SNPs) are marked with black crosses, and deletions are represented as red rectangles. A joint evaluation of all variants discovers that truth and query haplotypes are identical, despite variant representation differences. The truth VCF contains a 145 base deletion, whereas the query VCF contains a 47 base deletion, a 1 base deletion, and a 97 base deletion. By prematurely categorizing variants prior to evaluation into small and structural variants, this equivalence cannot be determined and variants would be classified as false positive (FP) and false negative (FN) variant calls instead of true positives (TP). Only by jointly evaluating the small and large deletions can vcfdist detect that such complex variants are in fact equivalent

**Table 1** Comparison of WhatsHap compare and vcfdist phasing evaluations relative to the Q100-dipcall truth VCF on the whole-genome GIAB-Q100 BED

| Dataset | Tool | Measured switch errors | Measured flip errors |
|---------|------|------------------------|----------------------|
| hifiasm-dipcall | WhatsHap | 610 | 396 |
|  | vcfdist | 494 | 390 |
| Q100-PAV | WhatsHap | 324 | 433 |
|  | vcfdist | 6 | 52 |
| hifiasm-GIAB-TR | WhatsHap | 1074 | 1004 |
|  | vcfdist | 494 | 396 |

WhatsHap consistently reports more switch and flip errors than vcfdist. We demonstrate in Additional File 1: Fig. S3 that most of these supposed phasing errors are actually correctly phased and provide an example in Fig. 4

variants improves measured performance. A similar example containing sequence information in VCF format is provided in Additional File 1: Fig. S2.

### Sophisticated variant comparison techniques result in better phasing evaluations

To understand joint phasing evaluation accuracy, we compare vcfdist to WhatsHap, a current standard for phasing evaluation [32, 33]. WhatsHap's compare module performs one-to-one variant comparisons between truth and query VCFs to evaluate phasing correctness. For each heterozygous query variant, WhatsHap searches for an identical truth variant and notes whether that truth variant has the same or opposite phasing of the corresponding query variant. Within each phase block, WhatsHap then uses a simple dynamic programming algorithm to minimize the total number of flip errors (in which the phase of a single variant is mismatched) and switch errors (in which the phases of all following variants are mismatched) [32]. Although WhatsHap's approach seems intuitively correct, it breaks down in repetitive regions of the genome where differing variant representations can result in false positive reported flip errors. Table 1 clearly shows that WhatsHap reports far more switch and flip errors than vcfdist on the exact same variant calls, particularly for the Q100-PAV and hifiasm-GIAB-TR datasets.

In contrast to WhatsHap, vcfdist performs full alignment of all nearby truth and query variants (a "supercluster") and is able to discover equivalencies in variant

representations. As a result, vcfdist reports far fewer phasing errors. The Q100-PAV VCF contains the fewest switch and flip errors, likely because it was produced using the same verkko assembly as the draft benchmark Q100-dipcall VCF. For the hifiasm-dipcall and hifiasm-GIAB-TR VCFs, vcfdist reports nearly identical switch and flip error rates. We believe this is because they were both produced using the same hifiasm scaffold [34]. In comparison, WhatsHap reports a much higher combined switch and flip error rate for the hifiasm-GIAB-TR VCF than for the hifiasm-dipcall VCF. We expect this is because the variant representation used by the hifiasm-GIAB-TR callset differs significantly from that used by the draft benchmark Q100-dipcall VCF, whereas the parameters used by the hifiasm-dipcall VCF are identical (Additional File 1: Fig. S1).

 In Additional File 1: Fig. S3, we present an extensive comparison of the switch and flip errors reported by WhatsHap and vcfdist. We find that 42.6% of flip errors reported by WhatsHap proved to be false positives, since the truth and query sequences match exactly when all neighboring variants are considered. An example is shown in Fig. 4, where WhatsHap reports a flip error within a complex variant even though both truth and query haplotypes match exactly. A further 49.6% of the flip errors reported by WhatsHap were not classified as flip errors by vcfdist due to insufficient evidence. Since there was no ground truth for these instances, we manually examined 16 random cases from each dataset (48 in total) where the ground truth was unknown. We found that classifying flip errors with WhatsHap resulted in 43 false positives, 4 true negatives, and 1 true positive. In comparison, classifying flip errors with vcfdist resulted in 40 true negatives, 7 false positives, and 1 false negative when compared to a manual examination. As can be seen in Table 1, these excess false positive flip error calls by WhatsHap artificially inflate the reported switch error rate as well, which will significantly impact tertiary analyses.

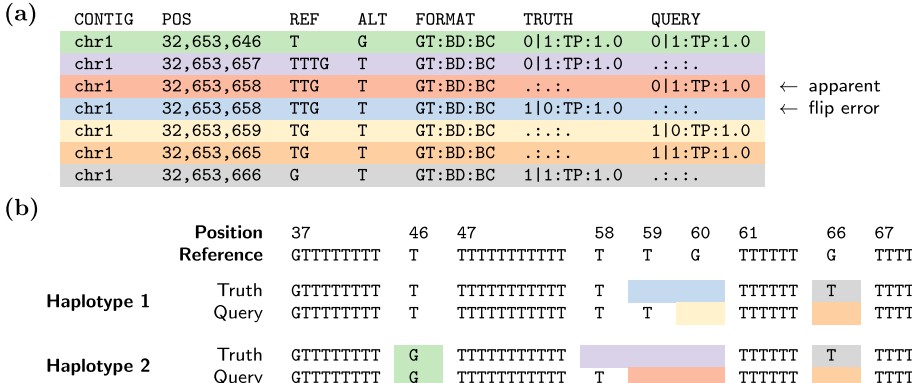

**Fig. 4  a** The variant call file (VCF) for an example WhatsHap false positive flip error call. Each VCF record shows the variant chromosome (`CONTIG`) and positions (`POS`) in addition to the reference (`REF`) and alternate (`ALT`) alleles and their genotypes (`GT`), the benchmarking decision (`BD`), and benchmarking credit (`BC`). In isolation, the two-base deletions at position 32,653,659 (the red and blue variants) appear to be the same variant (because `POS`, `REF`, and `ALT` match) phased differently between the truth and query VCFs (i.e., a flip error). **b** The resulting haplotype sequences. When this supposed flip error is considered in the context of the surrounding variants, vcfdist is able to determine that the two sets of truth and query variant calls are equivalent because both truth and query haplotype sequences are exactly the same. Both truth haplotypes contain an extra `T` at position 32,653,666 and one fewer `T` at positions 32,653,658-9. As a result, it is clear that no such flip error has occurred and differences between the truth and query VCF are due solely to differing variant representations

## vcfdist enables highly accurate comparisons with reasonable runtime

Next, we compare vcfdist to previously published works vcfeval [15] and Truvari bench [16], designed for evaluating small and structural variants, respectively. We also benchmark the performance of Truvari's refine module, a recently developed extension which realigns truth and query variants to one another using MAFFT [29], wavefront alignment (WFA) [28], or partial-order alignment (POA) [30] for more accurate benchmarking. Truvari refine achieves similar accuracy to vcfdist but changes the total counts of truth and query variants, making comparisons across different evaluation tools and pipelines difficult (see Table 3 for an example). All current versions of Truvari do not evaluate SNP accuracy, since Truvari was designed for SV evaluation.

At the other end of the spectrum, vcfeval only evaluates variants smaller than 1000 bases. For this reason, we restrict the maximum variant size to 1Kb in Fig. 5. As variant length increases in Fig. 5, vcfeval reports an increasingly high error rate compared to vcfdist (90.6% higher for SNPs, 128% higher for INDELs, and 321% higher for SVs). This is because vcfeval requires truth and query variants to match exactly (which is less likely for larger variants), whereas vcfdist and Truvari do not. A more lenient matching heuristic will lead to strictly fewer false positives and false negatives in Fig. 5. To avoid falsely inflating vcfdist's performance, we set vcfdist's credit threshold to 70% in order to match Truvari's sequence similarity threshold of 70% as closely as possible. We additionally standardize the method of variant counting across all tools in Fig. 5 to be consistent with vcfdist because otherwise differences in counting credit for partial allele matches would dominate the results (see Table 4).

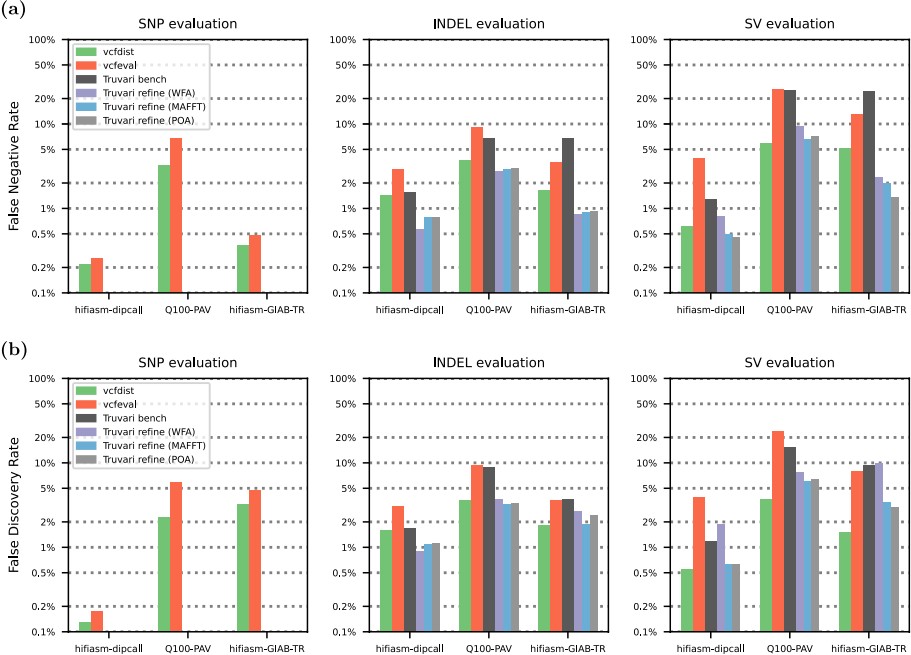

**Fig. 5** Comparison of vcfdist with prior works vcfeval and Truvari bench (and its unpublished `refine` variants) in terms of measured **a** false negative rate (FNR) and **b** false discovery rate (FDR) on the GIAB-Q100 BED, which contains benchmarking regions covering 90.3% of the human genome. Note that Truvari does not evaluate SNP accuracy, and that all results are plotted on a logarithmic scale

Figure 5 shows that vcfdist measures lower false negative and discovery rates for all variant sizes across all three datasets when compared to previously published works Truvari bench and vcfeval. It is able to accomplish this by allowing inexact matches, evaluating groups of variants simultaneously, and allowing variant matches to occur across size categories. In comparison to Truvari refine, vcfdist achieves a similar improvement in benchmarking accuracy but without modifying the variant representations during benchmarking, and evaluating SNPs in addition to INDELs and SVs. Despite vcfdist's advantages, we find that Truvari refine currently scales better with variant size, since it uses more memory-efficient alignment algorithms. We plan to incorporate wavefront alignment into the next release of vcfdist, but for now the maximum recommended variant length is 10 Kb. For more details on the advantages of WFA, see [28, 35]. Table 2 shows the runtimes of vcfdist, vcfeval, Truvari bench, and Truvari refine on our server; configuration details are provided in the "Methods" section.

Both Truvari and vcfdist perform alignment-based evaluation, which allows detection of variant calls that are mostly but not exactly correct. By default, Truvari considers a variant to be a true positive when the sequence similarity (percent of matching bases divided by total sequence length) exceeds 70%. We set vcfdist's default credit threshold (minimum percent reduction in edit distance when the reported variant is present) to 70% to match this. In practice, this is only slightly more stringent than Truvari's criterion (see "Different thresholds" in Table 4). This is crucial for identifying large structural variants that are less likely to be called perfectly. In contrast, vcfeval finds matching subsets of truth and query variants that result in the exact same haplotype sequence. This computation would be less expensive if not for the fact that vcfeval does not assume the input VCFs are phased. Because there are $2^n$ possible phasings for $n$ heterozygous variants, vcfeval's runtime depends more closely on the number and representation of variants than either Truvari or vcfdist. As a result, vcfeval has a wide range of runtimes. It is also important to note that when the number of nearby heterozygous variants is too large, vcfeval fails to compare these variants. This happened for 9712 variants (0.20%) on the hifiasm-dipcall VCF, for 21,886 variants (0.45%) on the Q100-PAV VCF, and for 136,073 variants (2.50%) on the hifiasm-GIAB-TR VCF.

The runtimes of vcfdist and Truvari, on the other hand, depend closely on the size of the sequences to be aligned. Truvari reduces total runtime by evaluating variants in two stages; only complex regions are passed on to Truvari's refine module for more sophisticated evaluation. vcfdist segments contigs into independent superclusters using

**Table 2** Runtime results for vcfdist, vcfeval, Truvari bench, and Truvari refine in *(h:)mm:ss* format

| | GIAB-Q100 BED Runtime | | |
| --- | --- | --- | --- |
| | hifiasm-dipcall | Q100-PAV | hifiasm-GIAB-TR |
| **vcfdist** | 52:19 | 1:04:56 | 59:22 |
| **vcfeval** | 14:40 | 50:39 | 46:18 |
| **Truvari bench** | 10:03 | 10:59 | 10:46 |
| **Truvari refine (MAFFT)** | 14:10 | 23:08 | 22:36 |
| **Truvari refine (WFA)** | 12:26 | 1:13:53 | 1:24:03 |
| **Truvari refine (POA)** | 11:57 | 21:47 | 24:39 |

**Table 3** Comparison of tools evaluating the `chr6:32,664,600-32,664,899` region of the HLA-DQB1 gene

| | | SNP results | | | | INDEL results | | | | SV results | | | | Total | |
| | | Truth | | Query | | Truth | | Query | | Truth | | Query | | | |
| | | TP | FN | TP | FP | TP | FN | TP | FP | TP | FN | TP | FP | FN | FP |
|---|---|---|---|---|---|---|---|---|---|---|---|---|---|---|---|
| vcfdist | hifiasm-dipcall | 32 | 0 | 32 | 0 | 2 | 0 | 2 | 0 | 2 | 0 | 2 | 0 | 0 | 0 |
| | Q100-PAV | 32 | 0 | 66 | 0 | 2 | 0 | 3 | 0 | 2 | 0 | 0 | 0 | 0 | 0 |
| | hifiasm-GIAB-TR | 32 | 0 | 71 | 0 | 2 | 0 | 1 | 0 | 2 | 0 | 0 | 0 | 0 | 0 |
| vcfeval | hifiasm-dipcall | 14 | 0 | 14 | 0 | 2 | 0 | 2 | 0 | 2 | 0 | 2 | 0 | 0 | 0 |
| | Q100-PAV | 13 | 0 | 13 | 4 | 1 | 0 | 0 | 0 | 2 | 0 | 0 | 0 | 0 | 4 |
| | hifiasm-GIAB-TR | 12 | 0 | 12 | 4 | 0 | 0 | 0 | 0 | 2 | 0 | 0 | 0 | 0 | 4 |
| Truvari bench | hifiasm-dipcall | * | * | * | * | 2 | 0 | 2 | 0 | 2 | 0 | 2 | 0 | 0 | 0 |
| | Q100-PAV | * | * | * | * | 2 | 0 | 2 | 1 | 0 | 2 | 0 | 0 | 2 | 1 |
| | hifiasm-GIAB-TR | * | * | * | * | 1 | 1 | 1 | 0 | 0 | 2 | 0 | 0 | 3 | 0 |
| Truvari refine | hifiasm-dipcall | * | * | * | * | 2 | 0 | 2 | 0 | 2 | 0 | 2 | 0 | 0 | 0 |
| | Q100-PAV | * | * | * | * | 3 | 0 | 3 | 0 | 0 | 0 | 0 | 0 | 0 | 0 |
| | hifiasm-GIAB-TR | * | * | * | * | 3 | 0 | 3 | 0 | 0 | 0 | 0 | 0 | 0 | 0 |

All three query VCFs called both haplotype sequences exactly correct; the resulting sequences are shared in Additional File 1: Fig. S4. As a result, there should be no false negative (FN) or false positive (FP) variant calls counted, and the number of truth VCF true positives (TP) should be consistent across all three datasets. The only tool that correctly ascertains this is vcfdist. SNP results for Truvari are marked with a * because Truvari does not evaluate SNPs

**Table 4** A comparison of INDEL and SV variant calling evaluation by vcfdist, vcfeval, and Truvari, restricting to chr20 of the GIAB-TR tandem repeats BED

| Category | Count | Allele match | Different thresholds | Complex variant | Pick single | Flip error | Backtracking tie | Variant overlap |
|---|---|---|---|---|---|---|---|---|
| **all agree FP** | 451 | | | | | | | |
| only vcfeval calls TP | 5 | | | | | | | 5 |
| only Truvari calls TP | 256 | 223 | 12 | 12 | | | | 9 |
| only vcfdist calls TP | 23 | | | 10 | 13 | | | |
| **all agree TP** | 35,376 | | | | | | | |
| only vcfeval calls FP | 662 | 346 | 307 | | | 1 | 8 | |
| only Truvari calls FP | 1 | | | | | | 1 | |
| only vcfdist calls FP | 52 | | 2 | 37 | | 6 | 6 | 1 |

SNPs were excluded because they were not evaluated by Truvari. Prior to evaluation, truth and query VCFs were normalized using Truvari phab. This means that the Truvari results reported in this figure are largely equivalent to Truvari refine (MAFFT). Variants that were not evaluated uniformly by the three tools are categorized and counted. Each category is described in greater detail within the manuscript text and an example provided in Additional File 1: Fig. S6

heuristics or a bidirectional wavefront algorithm (biWFA) [36], as shown in Additional File 1: Table S2.

### An example of benchmarking interpretability for the HLA-DQB1 gene

The results of variant call benchmarking tools such as vcfeval, Truvari, and vcfdist are frequently used in downstream tertiary analyses. In order to compare the interpretability of these tools, we evaluated the `chr6:32,664,600–32,664,899` region of the HLA-DQB1 gene, since it is known to cause difficulties during analysis arising from differing variant representations. The HLA-DQB1 gene is part of a family of genes called the human leukocyte antigen (HLA) complex and plays an essential role in the human immune system. Deleterious mutations in HLA-DQB1 are highly associated with common autoimmune diseases such as celiac disease [37] and multiple sclerosis (MS) [38].

We found through a manual examination that all three query VCFs called both haplotype sequences in this region exactly correct. A summary of these results is shown in Table 3, and the resulting sequences are included in Additional File 1: Fig. S4. Although all sequences were the same, there were significant differences in how this genetic variation was represented. Compared to the first Q100-dipcall haplotype, the Q100-PAV and hifiasm-GIAB-TR VCFs chose to represent a 169-base insertion and a 168-base deletion as a 1-base insertion and 34 SNPs. Relative to the second Q100-dipcall haplotype, the hifiasm-GIAB-TR VCF chose to represent a 1-base insertion and 1-base deletion as 5 SNPs. These differences account for the wide range of query true positive variant counts in Table 3.

Note that because both truth sequences were called exactly correct, there should be no false negative (FN) or false positive (FP) variant calls counted, and the number of truth VCF true positives (TP) should be consistent across all three datasets. vcfdist correctly counts all variants, as expected. vcfeval correctly counts the SVs but fails to evaluate a large number of the SNP and INDEL variants because there are too many heterozygous variants in close proximity. It excludes these variants from the analysis and proceeds with a warning that the region is too complex to be evaluated. Truvari bench does not evaluate the SNPs and fails to identify several of the INDELs and SVs. This failure occurs because it discards the SNPs prior to evaluation and therefore does not discover the two cases where numerous SNPs are equivalent to an insertion and deletion. Truvari refine also does not evaluate SNPs. It is able to detect that all variant calls are correct, though it does convert both query and truth SVs to an INDEL for the Q100-PAV and hifiasm-GIAB-TR datasets.

### Validation of vcfdist

Lastly, we compare vcfdist to existing variant calling evaluation tools in order to verify its correctness. Following variant normalization (described in the "Methods" section), we organize all variants evaluated by vcfdist, vcfeval, and Truvari in Table 4. 57,865 SNPs in this region are excluded from Table 4 because they were not evaluated by Truvari; we include these results in Additional File 1: Fig. S5a. An additional 723 INDELs and SVs occurring at the border of the GIAB-TR BED regions are excluded because they were only analyzed by some of the VCF comparison tools. Variants are then categorized based on the apparent reason for differences in evaluated correctness between the three

tools and counted. An example from each of the seven discovered categories is shown in Additional File 1: Fig. S6 and described in further detail below.

Firstly, all three tools handle allele matches differently, which accounts for the majority of differences in Table 4. Truvari will match a query homozygous variant to a truth heterozygous variant and consider both to be true positives. vcfeval will perform the same match but consider the variants to be a false positive and false negative. vcfdist will match the heterozygous variant to one haplotype of the homozygous variant, consider both to be true positives, and then consider the second haplotype of the query homozygous variant to be a false positive. None of these methods is best; rather, each has strengths for certain applications. We caution that users consider these performance differences based on their downstream goals.

The second most common area of disagreement between tools stems from the fact that they have different thresholds for considering variant calls to be true positives. For example, vcfeval requires variants to match exactly, whereas vcfdist requires variants to have a partial credit score above 0.7 and Truvari requires a sequence similarity above 0.7. For certain edge cases, such as where a length three deletion is called length four, even Truvari and vcfdist may differ.

The next most common differences are intentional implementation differences between the tools. In particular, vcfdist refuses to split a complex variant into multiple variants and consider only a subset of those to be correct. Truvari, by default, only allows a variant to participate in one match (with the `--pick single` parameter), regardless of allele count. vcfeval is the only tool that does not enforce local phasing and allows flip errors of nearby variants to occur.

Lastly, there are rare cases where unintentional implementation differences lead to slightly differing results. Not all backtracking algorithms behave identically, leading to cases in which different algorithms will adjudicate which of a pair of variants is a false positive differently. There are also differences due to directly adjacent or overlapping variants. For example, only vcfeval allows two separate insertions at the same exact location on the same haplotype and Truvari evaluates spanning deletions, whereas vcfeval and vcfdist do not.

## Discussion

In this paper, we demonstrate that evaluating small and structural variants together is necessary for discovering equivalent sets of truth and query variant calls. Furthermore, we show that intelligent variant comparison, which is able to identify equivalent variant representations, is important for accurate phasing analyses. We then show that vcfdist is now able to scale to whole-genome analysis of phased SNPs, INDELs, and SVs with improved accuracy over prior work. Lastly, we describe and explain the differences between vcfdist and prior work as they relate to our variant calling benchmarking results.

As variant calling performance improves and increasingly complex clusters of variants are evaluated, minor differences in the implementations of evaluation tools such as Truvari, vcfeval, and vcfdist begin to significantly impact the results. Currently, the way each tool handles partially correct variant calls differs greatly. Partial correctness can occur in many ways: a single insertion is called mostly but not entirely correct, a

homozygous variant is called heterozygous (genotype error), a deletion is called with the incorrect length, a heterozygous variant is called on the wrong haplotype (flip error), or only a subset of several variants that comprise a complex variant are called. In Table 4, we show that differences in how these cases are handled lead to significant differences in the reported summary metrics (Fig. 5). Ideally, as a community we would define a standard methodology to handle each of these cases and clearly delineate how to count and categorize these errors. While the GA4GH Benchmarking Team set an initial standard for counting small variant genotype errors as both FPs and FNs, and report a separate metric for number of genotype errors, we will need new standards for complex small and structural variants [26]. In the absence of a standardized approach, it is important that users of variant calling evaluation tools understand how each tool handles these cases, and the impact that may have on their results.

In this work, we have revisited some of the earlier design decisions we made in vcfdist v1. Although we still believe that total alignment distance is a useful supplementary metric to precision and recall curves, we now skip this computation by default, and allow re-enabling it with the −−distance parameter. We believe that stratifying precision and recall curves by variant size offers many of the same benefits, with a more easily interpretable result. We have also replaced variant calling partial credit with a credit threshold −−credit-threshold [0.7]. Partial credit is still calculated, but rather than assigning partial false and true positives (which is unnecessarily complicated and non-intuitive), we allow the user to select a partial credit threshold above which variants are considered true positives and below which variants are considered false positives. This more closely aligns with the behavior of other structural variant calling benchmarking tools such as Truvari. Lastly, vcfdist no longer realigns truth and query variants to a standard normalized representation by default. We found this behavior to be undesirable in Truvari refine because it complicates comparisons with other pipelines or datasets. vcfdist still retains this capability, however, which can be enabled using −−realign-query and −−realign-truth. Although we no longer enable this feature by default, we urge individuals benchmarking variant calling pipelines to be aware of the variant representations used in their truth and query VCFs.

At the moment, vcfdist is designed to compare phased variants from a single sample query VCF to a truth VCF. We plan to extend vcfdist in the near future to handle unphased variant calls as well, since many genomic datasets do not contain phasing information. Along a similar vein, we would like for vcfdist to be able to work with multi-sample population VCFs for use in genome wide association studies (GWAS). We believe joint evaluation of small and structural variants will be incredibly valuable in this context. In order to make this a reality, we will need to continue to improve the efficiency of vcfdist. The alignment-based calculation of precision and recall will need to be shifted to a wavefront alignment based implementation [28], and when large or many nearby variants are present, we may need to sacrifice accuracy in order to improve efficiency.

## Conclusions

Recent improvements in long-read sequencing accuracy have enabled calling phased small and structural variants from a single analysis pipeline. Despite this, the current standard tools for variant calling evaluation are only designed for either small (vcfeval)

or large (Truvari) variants. In this work we extend vcfdist——previously a small variant calling evaluator——to evaluate structural variants, making it the first benchmarking tool to jointly evaluate phased single-nucleotide polymorphisms (SNPs), small insertions/deletions (INDELs), and structural variants (SVs) for the whole genome.

We find that a joint evaluation reduces measured false negative and false positive variant calls across the board: by 28.9% for SNPs, 19.3% for INDELs, and 52.4% for SVs over 50 bases. Building on vcfdist's alignment-based evaluation, we also jointly analyze phasing accuracy. vcfdist identifies that 43% to 92% of all flip errors called by standard phasing evaluation tool WhatsHap are false positives due to differences in variant representations. Lastly, we compare the accuracy of vcfdist to prior works and demonstrate that it is able to find more true positive variant matches than vcfeval and Truvari bench. We find that vcfdist performs similarly to Truvari refine, while also providing more easily interpretable results.

## Methods

All scripts described below are available in the GitHub repository https://github.com/TimD1/vcfdist [39] in the `analysis-v2/` subdirectory.

### Datasets

#### Q100-dipcall VCF

The v0.9 Q100-dipcall draft benchmark VCF and its associated GIAB-Q100 BED containing small and structural variants were used as the ground truth VCF throughout this manuscript [13]. They were created during a collaboration between the Telomere-to-Telomere Consortium (T2T, https://sites.google.com/ucsc.edu/t2tworkinggroup/home), the Human Pangenome Reference Consortium (HPRC, https://humanpangenome.org/), and the Genome in a Bottle Consortium (GIAB, https://www.nist.gov/programs-projects/genome-bottle) in an attempt to establish a diploid whole genome benchmark that is perfectly accurate. The term "Q100" refers to a Phred quality score [40] of 100 or one error per ten billion bases (i.e., zero expected errors per human genome). The v0.9 draft benchmark contains many errors, but improvements are still being made towards this ultimate goal. A combination of data from Oxford Nanopore Technologies (ONT), Pacific Biosciences high-fidelity sequencing (HiFi), Strand-Seq, and Hi-C were used in combination with the trio-based verkko assembler [13] and manual review to create a high-quality assembly. Lastly, dipcall [41] was used to generate a VCF of this assembly relative to the GRCh38 reference FASTA.

#### Q100-PAV VCF

The same verkko assembly [13] was then used by researchers at the National Institute of Standards and Technology (NIST, https://nist.gov) to generate a second VCF using the Phased Assembly Variant Caller (PAV) [42]. For this reason, the Q100-PAV variant phasings match very closely with the Q100-dipcall phasings, as can be seen in Table 1. Note that by default, PAV merges some non-identical haplotypes, resulting in inexact variant calls.

### *hifiasm-dipcall VCF*

The hifiasm-dipcall VCF was created by the HPRC using a combination of ONT ultra-long (UL), HiFi, Hi-C, and Bionano optical mapping data [34]. First, the trio-based hifiasm assembler [43] was used to create the initial assembly using HiFi and Hi-C data. Bionano optical mapping data was used to verify these scaffolds, and a combination of manual variant curation and polishing with ONT-UL data was used to generate the final assembly [34]. Lastly, dipcall [41] was used to generate a VCF of this assembly relative to the GRCh38 reference FASTA.

### *hifiasm-GIAB-TR VCF*

The v4.20 hifiasm-GIAB-TR VCF and its associated GIAB-TR BED were generated by the Genome In A Bottle Consortium (GIAB) using the same hifiasm assembly, in addition to custom scripts that use minimap2 and paftools [44]. The methodology is described in detail in [31] and was part of an effort to create a high-quality tandem repeat benchmark. Because the same hifiasm assembly was used, the phasing analysis results of hifiasm-GIAB-TR closely match hifiasm-dipcall in Table 1, but the variant representation is much different than the other VCFs, as shown in Additional File 1: Fig. S1 (see https://github.com/ACEnglish/adotto/discussions/4 for details).

### Preprocessing

Prior to evaluation, multi-allelic variants were split using bcftools norm v1.17 [45] with the -m-any parameter. Where required, the HG002 sample was extracted from the original VCF into a single-sample VCF using bcftools query.

### Separate vs. joint evaluation of small and structural variants

For each dataset, each subset (small variants (SNPs and INDELs) only, structural variants only, and all variants) of variants was first extracted into a separate VCF using bcftools view v1.17 [45]. SVs were defined as insertions or deletions greater than or equal to 50 base pairs in size. INDELs are below this size threshold. vcfdist v2.5.0 [46] was used to compare each VCF to the Q100-dipcall ground truth within the GIAB-Q100 benchmarking regions. Scripts available in the small_sv_all/ directory of our GitHub repository [39] were used to calculate and plot the false negative and false discovery rates for each variant categorization within each VCF. The results are shown in Fig. 2.

### Description of vcfdist phasing analysis

The original vcfdist v1 release contained an experimental phasing analysis algorithm that was untested and unready for production. In this work, we extended the original algorithm, described in [17], to perform a proper evaluation of phasing. First, we added support for phase blocks using the input VCFs' FORMAT:PS fields. Unlike most other tools, vcfdist allows the ground truth VCF to contain phase sets as well. Using the reported phase sets, vcfdist now correctly identifies switch and flip errors. In addition to reporting detailed switch and flip error information, vcfdist also calculates several useful summary metrics such as phase block NG50 (breaking regions on new phase blocks), switch NGC50 (breaking regions on new phase blocks and switch errors), and switchflip

NGC50 (breaking regions on new phase blocks, switch errors, and flip errors). The NG50 metric reports the largest region such that all regions of that size and larger cover at least 50% of the genome. Lastly, we added a phasing threshold so that variant clusters are considered unphased unless one phasing significantly improves the cluster's edit distance from the ground truth versus the other phasing. The default value is set at ⸺ `phasing-threshold [0.6]`, or 60% reduction in edit distance, although the results from Additional File 1: Fig. S3c suggest that a higher threshold may be appropriate.

### Phasing analysis comparison with WhatsHap

First, because WhatsHap does not allow providing a BED file to mask analysis regions, we use bcftools filter v1.7 [45] to restrict all three VCFs to the GIAB-Q100 benchmarking BED regions. We then performed phasing analyses using WhatsHap v2.1 [32] and vcfdist v2.5.0 [46]. We then used several scripts, available in the `phasing/` directory of our GitHub repository [39], to compare and plot the resulting flip and switch errors reported by each tool. The results are shown in Additional File 1: Fig. S3 and Table 1. In Additional File 1: Fig. S3, a random subset of 16 clusters with unknown phasing from each VCF was selected and manually examined in order to define a ground truth and compare vcfdist's and WhatsHap's performances on this subset of cases.

### Accuracy comparison of variant calling evaluation tools

In order to determine the differences in variant evaluation between vcfdist, vcfeval, Truvari bench, and Truvari refine, we evaluated all three datasets using each tool on the GIAB-Q100 BED. We ran vcfdist v2.5.0 [17] with `-l 1000` to limit the maximum SV length because vcfeval does not consider variants larger than 1000 bases. We ran rtg vcfeval v3.12.1 [15] with the following parameters: ⸺`-ref-overlap` ⸺`-all-records` ⸺`-vcf-score-field=QUAL`. We ran Truvari bench v4.2.1 with the following command line options: ⸺`-no-ref a` ⸺`-sizemin 1` ⸺`-sizefilt 1` ⸺`-sizemax 1000` ⸺`-pick single` ⸺`-typeignore` ⸺`-dup-to-ins`. We then ran Truvari refine v4.2.1 using ⸺`-regions candidate.refine.bed` from the previous Truvari bench step with the following parameters: ⸺`-use-original-vcfs` ⸺`-use-region-coords` ⸺`-recount` ⸺`-align <method>` where `<method>` was WFA, MAFFT, and POA. We then merged results from Truvari refine and bench by subtracting the `candidate.refine.bed` (used by Truvari refine) from the GIAB-Q100 BED using bcftools subtract v1.17, evaluating the remaining regions with Truvari bench, and then merging the results. Lastly, we used several scripts, available in the `vs_prior_work/` directory of our GitHub repository [39], to compare and plot the false negative and false discovery rates of each tool. These results are shown in Fig. 5.

The same methodology was used to evaluate the example complex variant in the HLA-DQB1 gene for each tool. Scripts are available in the `hla/` directory of our GitHub repository [39], and results are shown in Table 3.

### Validation of vcfdist

In order to validate the correctness of vcfdist's reported variant calling results, we needed a way to compare how vcfdist classified a single variant to the classifications reported by prior works vcfeval and Truvari refine (MAFFT) for that same variant. We chose to

compare against Truvari refine instead of Truvari bench because it is more accurate [31]. Unfortunately, Truvari refine changes the representations of the input truth and query variants, making it difficult to compare benchmarking results across tools. Instead of directly using Truvari refine, we used an equivalent workflow that involves normalizing variant representations using Truvari phab (which internally uses MAFFT) followed by Truvari bench evaluation. This approach enabled us to directly compare the decisions made by vcfdist, vcfeval, and Truvari on the exact same set of variants while also enabling Truvari to be more accurate.

We first normalized chr20 of the GIAB-TR BED from each of our three VCFs with the truth VCF using Truvari v4.2.1 phab (MAFFT). We then converted these VCFs into single sample VCFs of the desired format and split up multi-allelic variants using bcftools reheader, norm, and view v1.18 [45]. Afterwards, we evaluated each VCF using vcfdist v2.5.0 [17], vcfeval v3.12.1 [15], and Truvari bench v4.2.1 [16]. Scripts available in the `phab_cm/` directory of our GitHub repository [39] were used to run and summarize the evaluations. The results are presented in Table 4, Additional File 1: Fig. S5, and Additional File 1: Fig. S6.

### Improvements to the vcfdist clustering algorithm

A naive variant comparison algorithm would compare each query variant to a single reference variant individually in order to discover matches. While this approach works for the majority of variant calls, there are cases where several query variants are equivalent to one or many truth variants (Additional File 1: Fig. S7a). Several examples of this are shown in Additional File 1: Fig. S8. This is especially true for repetitive regions of the genome, or as the representations of the truth and query VCFs diverge. In order for a benchmarking tool to recognize these cases of complex equivalency, all the variants involved must be evaluated at once in a group or "cluster."

The default clustering algorithm employed by vcfdist discovers all cases in which variants could participate in a complex match and groups those variants together into a single cluster. vcfdist achieves this by first initializing each cluster to a single variant. Next, the leftmost and rightmost reference positions that can be reached by aligning through each cluster with an alignment cost less than or equal to the cost of the current variant representation are recorded. If the reach of a cluster overlaps with the reach of a neighboring cluster, the two clusters are merged. This occurs until all clusters have stabilized.

In order to handle structural variants, vcfdist's original clustering algorithm (described briefly above and in greater detail in [17]) was required to undergo significant changes to improve efficiency and reduce memory usage. Firstly, the bidirectional Smith-Waterman-Gotoh [47] algorithm used to calculate cluster left and right reaches was converted to a wavefront alignment based [28] equivalent in order to reduce memory usage from $O(n^2)$ to $O(n)$. Next, the alignment cost of each cluster was recalculated (and lowered, when possible) following each cluster merge in order to reduce unnecessary cluster growth in following iterations. The left and right reaches of each cluster were cached across iterations and only recalculated following a merge. A greedy merging strategy was employed to merge multiple clusters at once when possible. Cluster reaches were calculated using iterative reach doubling to avoid unnecessary computation. Lastly, multithreading support was added for clustering.

## Comparison of vcfdist clustering methods

vcfdist v2.5.0 was run with "gap $n$" clustering for $n = (10, 100, 500)$ in addition to the default biWFA clustering using the command line options --cluster gap 10. Wall clock runtime was measured using the GNU time command on an Intel Xeon E5-2697 v3 CPU, with 56 threads and 64GB RAM. Scripts used to calculate the cluster sizes shown in Additional File 1: Table S2 from vcfdist's verbose outputs are available in the clustering/ directory of our GitHub repository [39].

## Extending vcfdist to evaluate structural variants

In order to evaluate larger structural variants using vcfdist, we made several changes such as introducing command line parameter --largest [5000] to control the size of variants evaluated by vcfdist. We also added summary metric reporting for structural variants separately from INDELs and added the --sv-threshold [50] flag to control the threshold for this classification. In addition to the numerous clustering efficiency improvements mentioned above, we decreased the memory usage of the precision and recall calculations. Since each cluster can be evaluated independently, we also added multi-threading and work balancing based on cluster size for all intra-cluster computations.

## Runtime comparison of variant calling evaluation tools

Variants larger than 1000 bp were pre-filtered using bcftools view v1.17 and inversions were filtered using GNU grep v2.20 because otherwise they significantly impact the runtime of Truvari refine (even when excluded from the analysis with --size-max 1000). Wall clock runtime was measured using the GNU time command on an Intel Xeon E5-2697 v3 CPU, with 56 threads and 64GB RAM. The scripts with the exact parameters used to run each tool are available in the vs_prior_work/ directory of our GitHub repository [39]. The results are shown in Table 2.

## Supplementary information

---

Additional file 1. Contains the following tables and figures: Table S1: The origins of each phased whole genome sequencing dataset used in this manuscript. Figure S1: The design space for affine-gap alignment and variant representations. Figure S2: An example where joint evaluation of small and structural variants changes benchmarking results. Figure S3: Switch and flip error confusion matrices for vcfdist and WhatsHap. Figure S4: Full haplotype sequences for the truth and query VCFs for a portion of the HLA-DQB1 gene. Figure S5: A comparison of SNP, INDEL, and SV variant calling evaluation by vcfdist, vcfeval, and Truvari. Figure S6: Examples of the seven variant categories in Table 4 that were evaluated differently between tools. Table S2: Comparing the efficiency and accuracy of vcfdist when using different clustering approaches. Figure S7: Comparing detected and actual variant dependencies when using different clustering algorithms. Figure S8: An example of different variant clustering methods.

Additional file 2. Contains the peer review history.

---

### Acknowledgements

We thank Nathan Olson from NIST for his work on developing the Q100-dipcall draft benchmark dataset and Nathan Dwarshuis from NIST for generating the Q100-PAV dataset. We also thank Adam English for his feedback and suggestions on improving our evaluations of Truvari bench and refine. Lastly, we thank Vincent Laufer from the University of Michigan for his suggestions on improving the manuscript.

### Review history

The review history is available as Additional file 2.

**Peer review information**

**Authors' contributions**

T. D. developed vcfdist and performed the computational studies. J. M. Z. suggested and guided the project. T. D. and J. M. H. worked on the phasing analysis. S. N. supervised the work. All authors contributed to the final paper and figures.

**Funding**

This material is based upon work supported by the National Science Foundation under Grants No. 1841052 (T. D.) and 2030454 (S. N.). Any opinion, findings, and conclusions or recommendations expressed in this material are those of the authors and do not necessarily reflect the view of the National Science Foundation. This work was also supported by the Kahn Foundation (S. N.).
 Certain commercial equipment, instruments, or materials are identified to specify adequately experimental conditions or reported results. Such identification does not imply recommendation or endorsement by the National Institute of Standards and Technology nor does it imply that the equipment, instruments, or materials identified are necessarily the best available for the purpose.

**Availability of data and materials**

All input data for this manuscript have been deposited in a Zenodo repository, publicly available at the following URL: https://doi.org/10.5281/zenodo.10557082 [48]. The deposited data includes the Q100-dipcall VCF, Q100-PAV VCF, hifiasm-dipcall VCF, hifiasm-GIAB-TR VCF, GIAB-Q100 BED, GIAB-TR BED, and the GRCh38 reference FASTA.
 All code for vcfdist and the benchmarking pipelines developed for this manuscript are available in a public GitHub repository (https://github.com/TimD1/vcfdist) under a permissive GNU GPLv3 license [39]. The repository has also been deposited in Zenodo, at https://doi.org/10.5281/zenodo.8368282 [46].

## Declarations

**Ethics approval and consent to participate**

Ethical approval was not needed for this study.

**Consent for publication**

Not applicable.

**Competing interests**

J. M. H. is employed by and holds stock in PacBio.

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

## 