## [Additional file 2. Contains the peer review history. · Genome Biology]

Review history

First round of review

Reviewer 1

In the presented manuscript, Dunn and colleagues describe new software designed to compare the results of different variant calling tools. The authors convincingly demonstrate the main advantage of their software: it is more logical and accurate to jointly compare small (i.e., SNVs and indels) and large (i.e., SV) calls. This is particularly important in repetitive regions, where multiple variants and variant call representations are ambiguous, causing tools to report the same variants differently. Joint call comparison is warranted in these cases to ensure accurate variant and phase comparisons. The software's runtime characteristics are comparable to other similar tools.

While the software is useful, I find it to be of a narrow technical scope. It is specifically designed to compare other computational tools and, by itself, does not lead to any biological insights. Similarly, the manuscript is highly technical in nature and lacks biological context. I believe it would be better suited for a more specialized journal.

Additionally, I have the following reservations about the manuscript in its current form:

- * Figure 3 is unclear. The method for calculating counts of FP, TP, and FN is not well-explained. The "ALL" variants in Query and Truth seem different, which is confusing. I am puzzled by the presence of 5 TPs. This figure appears to be central to the manuscript's main message.
- * I struggled significantly to understand Figure 4. Most readers of Genome Biology might not grasp its content and this underscores the narrow and technical focus of the manuscript,
- * Citations are missing from the manuscript.
- * There are several typos and spelling errors in the text.

Reviewer 2

Dunn et al. propose a new tool for the benchmark of variant calling, named as *vcfdist*. The major contribution is that the tool enables to joint benchmark SNVs, indels and structure variations (SVs), which helps to reduce the false positives and negatives made by the various representations of equivalent variants. The tool is well-designed and solves several critical problems in variant calling benchmark. I believe that it may play important roles in many genomics studies. The manuscript is well-written and I think it will be acceptable if several issues are addressed as following.

1. The authors evaluated the results of the tool by several well-studied callsets and did a case study on HLA-DQB1 region. I think that more case studies could be more helpful to reveal the behaviors of state-of-the-art benchmark tools and their impacts to the benchmark. Especially, I suggest to focus on more regions with complex variants, such as SV hotspot regions.
2. The authors claimed their plan to use alignment-based approach to achieve better scalability for variant size. They suggest wavefront alignment. It could be better to give a more detailed discussion about the way to use alignment method here, i.e., talk about more about the design of the alignment approach, scoring parameter and their applicability to reduce the divergent representations of equivalent variants.
3. The authors stated that "Both Truvari and *vcfdist* perform alignment-based evaluation, which allows detection of variant calls that are mostly, but not exactly correct." (Page 8, last row.) This sentence is important and should be more informative. Is there a measure to quantify how different the calls are allowed for Truvari and *vcfdist* to make correct judgement?

4. From Table 2, I observed that the timecost of vcfdist is relatively high, especially for the hifiasm-dipcall dataset. More detailed discussions are helpful. What operations or steps are computational expensive? Is it scalable to larger datasets?

Authors' response to reviewers

Firstly, we thank the editor and both reviewers for their time and efforts in reviewing our manuscript. Below, we have provided a point-by-point response to all comments and concerns raised by both reviewers.

Responses to Reviewer #1:

In the presented manuscript, Dunn and colleagues describe new software designed to compare the results of different variant calling tools. The authors convincingly demonstrate the main advantage of their software: it is more logical and accurate to jointly compare small (i.e., SNVs and indels) and large (i.e., SV) calls. This is particularly important in repetitive regions, where multiple variants and variant call representations are ambiguous, causing tools to report the same variants differently. Joint call comparison is warranted in these cases to ensure accurate variant and phase comparisons. The software's runtime characteristics are comparable to other similar tools.

This is an excellent summary of our manuscript and contributions.

While the software is useful, I find it to be of a narrow technical scope. It is specifically designed to compare other computational tools and, by itself, does not lead to any biological insights. Similarly, the manuscript is highly technical in nature and lacks biological context. I believe it would be better suited for a more specialized journal.

The editors have stated that they do not share this concern (narrow in scope and lacking biological insight). Our manuscript is being considered as a Method article, whose criteria specify that it “need not necessarily provide novel biological insights” [1]. We agree that the content of the manuscript covers a rather technical scope, detailing the intricacies of variant representations and how different comparison methods and algorithms affect the results of variant call benchmarking. However, we believe that the impact of our contributions will not be of narrow scope. Our manuscript improves the accuracy of variant call benchmarking, which is necessary for evaluating novel sequencing technologies, methods, and variant callers. Our software is not intended to directly lead to new biological insights, but rather to enable optimal sequencing platform and tool selection by researchers and bioinformaticians, thus indirectly improving the accuracy of all resulting sequencing pipelines. Furthermore, we identify the (previously unknown) problem that simpler variant comparisons pose for accurate phasing evaluation: at least 42% of the flip “errors” reported by What-sHap were not actually errors!

Additionally, I have the following reservations about the manuscript in its current form:

1. Figure 3 is unclear. The method for calculating counts of FP, TP, and FN is not well-explained. The "ALL" variants in Query and Truth seem different, which is confusing. I am puzzled by the presence of 5 TPs. This figure appears to be central to the manuscript's main message.

We have added more annotations to Figure 3 to increase its readability, and expanded the caption to explain how the truth and query VCF contain the same complex variant represented in two different ways. We explain how small and structural variants must be evaluated jointly during benchmarking in order to discover such representation equivalencies.

2. I struggled significantly to understand Figure 4. Most readers of *Genome Biology* might not grasp its content and this underscores the narrow and technical focus of the manuscript.

This manuscript largely deals with complex variants, and so the examples we have included are unfortunately – by definition – complex. We understand that these examples can be difficult to parse, and so we have added coloring and improved the textual explanation of Figure 4.

Although the manuscript is technical, we do not believe that its technicality detracts from its importance. We found that across three WGS datasets, 42% of all phasing flip “errors” reported by existing phasing evaluation tool WhatsHap were similar to the example in Figure 4: they were incorrectly labelled as errors due to differences in complex variant representations. This suggests that while these complex cases are relatively few in number, they make up a significant portion of the falsely labeled “errors” reported by existing tools.

3. Citations are missing from the manuscript.

We have been unable to get the Genome Biology web server to correctly build and link our references, despite trying several different methods that each work locally. Our resubmitted manuscript includes a full PDF copy of our manuscript near the end (directly prior to this response letter), attached as a “Supplementary File” labelled *manuscript.pdf*. This compiled manuscript contains our correctly-embedded citations and bibliography.

4. There are several typos and spelling errors in the text.

We have carefully reviewed the entire article and not found any such typos or spelling errors. Perhaps you are referring to the text-based reference abbreviations that have been inserted into our manuscript submission instead of numeric references, caused by a malfunctioning L^AT_EX build (e.g. [*mongolism, sv-human*] instead of [1,2]). If there are other typos or spelling errors you have identified, we would be happy to fix them.

Responses to Reviewer #2:

Dunn et al. propose a new tool for the benchmark of variant calling, named as vcfdist. The major contribution is that the tool enables to joint benchmark SNVs, indels and structure variations (SVs), which helps to reduce the false positives and negatives made by the various representations of equivalent variants. The tool is well-designed and solves several critical problems in variant calling benchmark. I believe that it may play important roles in many genomics studies. The manuscript is well-written and I think it will be acceptable if several issues are addressed as following.

Thank you!

1. The authors evaluated the results of the tool by several well-studied callsets and did a case study on HLA-DQB1 region. I think that more case studies could be more helpful to reveal the behaviors of state-of-the-art benchmark tools and their impacts to the benchmark. Especially, I suggest to focus on more regions with complex variants, such as SV hotspot regions.

There is only a solid definition of “ground truth” for variant classification when the truth and query sequences match exactly. In this case, all variants should be classified as true positives. Our manuscript currently contains several examples of matching query and truth sequences but differing variant representations (Figure 3, Figure 4, and Supplementary Figure 2) in addition to the HLA-DQB1 case study you mention (Table 3 and Supplementary Figure 4), and so we do not believe that adding another similar case study would be beneficial.

Each of these case studies are complex to both evaluate and explain. Given the limited space available in this document, we have instead chosen to examine nearly a thousand cases where the behaviors of benchmarking tools differ, and identified seven different categories of reasons for why this occurs (Table 4). Those reasons are described in detail in

the manuscript (lines 226-258), and examples are provided in Supplementary Figure 6.

2. The authors claimed their plan to use alignment-based approach to achieve better scalability for variant size. They suggest wavefront alignment. It could be better to give a more detailed discussion about the way to use alignment method here, i.e., talk about more about the design of the alignment approach, scoring parameter and their applicability to reduce the divergent representations of equivalent variants.

In the interest of space, we do not go into details in the document as it is outside the scope of this paper. However, we have clarified that the reader can learn more about the algorithms by following specific references.

3. The authors stated that “Both Truvari and vcfdist perform alignment-based evaluation, which allows detection of variant calls that are mostly, but not exactly correct.” (Page 8, last row.) This sentence is important and should be more informative. Is there a measure to quantify how different the calls are allowed for Truvari and vcfdist to make correct judgement?

We have expanded upon this sentence, explaining that Truvari has a default sequence similarity threshold of 70% (percent of matching bases divided by total sequence length). We set vcfdist’s default threshold to 70% (percent reduction in edit distance when the variants are included) to match this. For edge cases, vcfdist is actually slightly more

stringent (see “Different Thresholds” in Table 4) than Truvari. Both tools allow changing the default value of this parameter, depending on the user’s application and the stringency that they require for a variant to be considered correct.

4. From Table 2, I observed that the time cost of vcfdist is relatively high, especially for the hifiasm-dipcall dataset. More detailed discussions are helpful.

The runtimes of vcfdist and Truvari are both relatively stable across all three WGS datasets. It is actually the fact that vcfeval’s runtime varies drastically (running much faster on the hifiasm-dipcall dataset because the variant representations are so similar) that vcfdist’s runtime appears to be an outlier. vcfeval is able to run much faster because it only considers exact matches that are equivalent, whereas vcfdist performs full sequence alignment and can find inexact matches. Of the three datasets, vcfdist actually runs the fastest on hifiasm-dipcall (Table 2).

What operations or steps are computational expensive?

The most computationally expensive component is performing alignment within each cluster as part of the precision and recall calculation. In future work, we expect to improve the runtime of this step by around 10×. This is achievable because the current implementation is unable to fully use the available parallelism. At the moment, vcfdist reduces the total threads when aligning large clusters (down to 1 thread for clusters of 10kb) due to memory limitations. By using WFA [2] for alignment and reducing memory usage, vcfdist will be able to run 64 threads in parallel when evaluating large clusters. Even in its current state of limited parallelism, vcfdist has only slightly slower runtimes than competing solutions.

Is it scalable to larger datasets?

These whole-genome sequencing datasets are the largest we would expect vcfdist to be run on, since vcfdist was designed for benchmarking human genomes and benchmarking is typically done with a single sample and ground truth genome. It certainly could be run on larger diploid genomes as well; runtime will be roughly proportional to the genome length.

References

- [1] BioMed Central Ltd. *Genome Biology: Method*. 2024. url: <https://genomebiology.biomedcentral.com/submission-guidelines/preparing-your-manuscript/method>.
- [2] Santiago Marco-Sola et al. “Fast gap-affine pairwise alignment using the wavefront algorithm”. In: *Bioinformatics* 37.4 (2021), pp. 456–463.